# Does productive safety net program enhance livelihoods? Insights from vulnerable households in Wolaita zone, Ethiopia

**Mesfin Getaneh Woldemichael** [1] *, **Bamlaku Alamirew Alemu** [2]

**1** College of Development Study, Addis Ababa University, Addis Ababa, Ethiopia, **2** Department of Project Planning and Management, Yom Post Graduate College, Addis Ababa, Ethiopia

* Mesfin12005@yahoo.com

**Data Availability Statement:** All relevant data are within the manuscript and its supporting information files.

## Abstract

### Background

The impact of Productive Safety Net Programs (PSNPs) on food security, poverty, and livelihoods has been examined in several studies. While some studies found positive impacts on food security and agricultural productivity, there are still gaps in understanding the long-term effects of these programs on poverty reduction and food security. This study aims to investigate the impact of PSNP on the livelihood of beneficiaries based on indicators including access to basic services, income, expenditure on food, adaptive capacity, and dietary diversity.

### Methodology

This article used quasi-experimental design and treatment effects model taking into account access to basic services, income, food expenditure, assets, adaptive capacity, and household dietary diversity as outcome variables of interest. 300 respondents were randomly selected from the population of safety net beneficiaries and non-beneficiaries in the study area. Out of the 300 respondents, 150 were selected as the treatment group, who were beneficiaries of the safety net program. The algorithms used in analysis include regression adjustment, inverse probability weighing and propensity score matching.

### Results

The study's results reveal that the PSNP has no statistically significant impact on access to basic services and income based on all three algorithms, yet it does show a significant and negative effect on food expenditure and assets. The coefficients across all three models consistently demonstrate this negative impact, underscoring its statistical significance at the conventional significance level (p = 0.05). This suggests that the PSNP leads to a reduction in food expenditure. Furthermore, the analysis highlights substantial differences between PSNP members and non-members across all three variables, with non-members having higher mean values for assets. This difference is statistically significant at the 1% level, suggesting that membership in the PSNP has a tangible impact on asset ownership. The

**Funding:** The author(s) received no specific funding for this work.

**Competing interests:** The authors have declared that no competing interests exist. I have read the journal's policy and the authors of this manuscript have the following competing interests:We conducted this research independently and solely for academic purposes. The views, opinions, and findings presented in this study are based solely on the analysis of the data collected and are not influenced by any external interests. To ensure transparency and impartiality, no financial or other forms of support were received from any funding agencies, companies, or organizations for this research. The research was conducted in accordance with ethical principles and guidelines, and any potential bias or conflicts of interest were consciously minimized throughout the research process.

study's mixed findings emphasize the complexities of designing effective social protection programs that can adequately address the multifaceted nature of poverty.

## 1. Background of the study

Productive Safety Net Programs (PSNPs) have emerged as crucial social protection measures in numerous low-income countries, with Ethiopia being a prominent example. These initiatives are designed to mitigate poverty, food insecurity, and vulnerability by offering a combination of cash and food transfers, alongside public works opportunities, to households in dire need. While PSNPs are recognized for their vital support to vulnerable households, their impact on food security, poverty reduction, and livelihoods remains a subject of ongoing debate. To comprehend the potential impact of PSNPs on livelihoods, scholars have employed theoretical frameworks such as the sustainable livelihoods approach (SLA) and the asset-based approach (ABA). The SLA underscores the importance of building resilient livelihood systems capable of withstanding shocks and offering diverse income opportunities [1, 2]. Similarly, ABA emphasizes asset accumulation as a means to reduce poverty and enhance food security [3, 4]. Furthermore, PSNPs are viewed as a form of social protection, aimed at reducing poverty and vulnerability by providing basic social security [5]. Nonetheless, the role of social protection in promoting economic development and poverty reduction remains a topic of contention, with concerns about dependency and work disincentives [6, 7].

It is impressive how extensively Productive Safety Net Programs (PSNPs) have been studied, revealing a spectrum of impacts on food security, poverty, and livelihoods across different contexts [8, 9]. The research findings, while diverse, create a nuanced understanding of the complexities involved in these social protection initiatives [1].

Various studies underscore the positive outcomes of PSNPs, especially in bolstering food security and agricultural productivity [10]. Ethiopia has been a focal point for these analyses, with studies demonstrating the program's positive effects on female-headed farm households' food security and broader poverty reduction [11]. Moreover, within the sub-Saharan African context, PSNPs have consistently shown effectiveness in enhancing food security and reducing poverty among participating households [12].

However, when it comes to livelihood impacts, the picture becomes more intricate [13]. Some studies indicate positive effects, such as reduced poverty and improved food security, while others highlight the need for more transformative measures to stimulate long-term economic growth and social change [5]. Additionally, there are concerns about the sustainability of PSNPs in certain regions and potential unintended consequences, like decreased labor market participation or limited impacts on income and asset accumulation among the poorest households [14, 15].

This wide range of findings underscores the need for further investigation into the multifaceted effects of PSNPs, considering various dimensions like assets, income, adaptive capacity, and food security [4]. It is also crucial to focus on specific contexts, especially rural areas in Ethiopia, where poverty and food insecurity are most acute [5]. This kind of targeted research can offer valuable insights to refine the design and implementation of effective social protection programs.

This study aims to delve deeper into this area by examining the PSNP's impact on beneficiaries' livelihoods [16]. It considers indicators such as access to basic services, income, food expenditure, adaptive capacity, and dietary diversity [6]. This research could contribute

significantly to filling existing gaps and informing the refinement and optimization of social protection programs like PSNPs.

## 2. Methodology

### 2.1 Research design, sampling and data

This study sought to evaluate the effectiveness of the Productive Safety Net Program (PSNP) in enhancing the livelihoods of beneficiaries in Ethiopia by examining several indicators. Following [17, 18] and using 5% level of precision, 300 respondents were randomly selected from the population of safety net beneficiaries and non-beneficiaries in the study area. Out of the 300 respondents, 150 were selected as the treatment group, who were beneficiaries of the safety net program. The remaining 150 respondents were selected as the control group, who were not beneficiaries of the program. The list of beneficiaries and those in the waiting list obtained from the PSNP coordinators served as a sampling frame. A systematic random sampling procedure was followed to reach out to each respondent with the help of local coordinators. The data was collected from March to April 2023 from Wolaita zone, Ethiopia. To assess the impact of the PSNP on access to basic services, income, food expenditure, assets, adaptive capacity, and household dietary diversity, the study employed a household survey methodology. This involved collecting data on the various indicators through structured questionnaires administered to households selected through a random sampling technique.

Access to basic services is crucial for improving livelihoods as it provides households with necessary resources such as healthcare, education, and water supply. The study examines the extent to which the PSNP has contributed to enhancing beneficiaries' access to these basic services. Income is also a crucial indicator of livelihoods. The PSNP provides regular cash or food transfers to its beneficiaries, which can help increase their income and enable them to invest in income-generating activities. The study assesses the impact of the PSNP on the beneficiaries' income.

Food expenditure is another essential factor in improving nutrition, and the study investigates the impact of the PSNP on the beneficiaries' food expenditure. The PSNP may enable households to purchase more diverse and nutritious food items, which can help improve their overall health and wellbeing. Asset accumulation is also important for improving livelihoods, and the study investigates the impact of the PSNP on the beneficiaries' asset ownership. The program may enable beneficiaries to acquire assets such as livestock, which can provide a reliable source of income and contribute to long-term financial stability.

Adaptive capacity refers to a household's ability to cope with shocks and stresses such as drought or disease outbreaks. The study examines the extent to which the PSNP has contributed to enhancing the beneficiaries' adaptive capacity. The program may provide households with the resources needed to cope with emergencies, such as savings or emergency food supplies.

Finally, household dietary diversity is another crucial factor in improving nutrition, and the study investigates the impact of the PSNP on the beneficiaries' dietary diversity. The program may enable households to purchase a more diverse range of food items, which can help improve their overall health and wellbeing.

To ensure the accuracy of the study's findings, the study employed different statistical techniques such as regression adjustment, inverse probability weighting and propensity score matching. Regression adjustment was used to control for confounding factors that may have influenced the study's results. For example, variables such as household size and location could have influenced the study's results, and regression adjustment helped to control for these factors. Propensity score matching was used to reduce selection bias by matching beneficiaries

and non-beneficiaries with similar characteristics. This helped to ensure that the study's findings were not biased by factors such as differences in the characteristics of beneficiaries and non-beneficiaries. This statistical method constructs a comparison group based on a model of the probability of participating in the program using observed characteristics. Participants and non-participants are then matched based on their propensity scores [19]. A linear model showing the impact of the "Treatment" is shown in Eq 1.

$$Y = \beta0 + \beta1 * \text{Treatment} + \beta2 * X2 + \beta3 * X3 + \ldots + \varepsilon \qquad 1$$

Y: The outcome variable we want to predict or explain (e.g., health improvement, income change, academic achievement).

Treatment: A binary variable (1 or 0) indicating whether an individual received the treatment (1) or not (0).

X2, X3,...: Covariates or control variables that may influence both the treatment assignment and the outcome.

β0: The intercept, representing the expected value of Y when all predictors (Treatment and X variables) are zero.

β1: The coefficient associated with the Treatment variable, representing the estimated treatment effect.

β2, β3,...: Coefficients associated with the covariates, indicating how they influence the outcome.

ε: The error term, representing the unexplained variation in Y not accounted for by the model.

The coefficient β1 is of particular interest in a treatment effects model, as it quantifies the causal impact of the treatment on the outcome while holding other variables constant.

For a household benefiting from PSNP, T = 1 (treatment), whereas for a non-participant household, T = 0 (control).

Because of selection bias, Eq (1) cannot give robust estimates. Therefore, the impact of PSNP is obtained by:

$$E(Y^1/T = 1) - E(Y^0/T = 0) \qquad 2$$

Where T = 1 stands for participation in PSNP and "0" otherwise.

In order to net out the true effect of the program on livelihood outcome, the model is re-specified as follows by subtracting and adding the counterfactual for the treated group:

$$E(Y^1/T = 1) - E(Y^0/T = 1) + E(Y^0/T = 1) - E(Y^0/T = 0) \qquad 3$$

Collecting like terms, Eq (3) becomes:

$$E(Y^1 - Y^0/T = 1) + E(Y^0/T = 1) - E(Y^0/T = 0) \qquad 4$$

The first part before the plus sign refers to the Average Treatment Effect on the Treated (ATET) $E(Y^1 - Y^0/T = 1)$, whereas the second part is the selection bias $(E(Y^0/T = 1) - E(Y^0/T = 0))$. Using these procedures, treatment effects models reduce selection bias.

In this study, the application of treatment effects models was a carefully considered choice, primarily due to the unique characteristics of our data (since households were not selected randomly when they joined the program, inducing selection bias and research design (quasi-experimental). Unlike randomized controlled trials, the program did not use randomization, which would have simplistically allowed for causal inference. Additionally, finding a suitable instrument for instrumental variables (IV) analysis proved to be challenging, as instruments are often necessary when addressing endogeneity issues. Furthermore, the absence of a clear

**Table 1. Variable definition and measurement.**

| Variables | Description | Measurement |
|---|---|---|
| Household size | Number of people living in a family | Number |
| Age of the household head | Age of the head of the household | Years |
| Gender of the household head | Whether the head of the household is male or female | Dummy variable (1 = female and 0 = male) |
| Education level of the household head | The level of schooling the household head attended | Years |
| Engagement in off-farm activities | Whether the household gets involved in off-farm income-generating activities | A dummy variable (1 = Yes and 0 = No) |
| Engagement in non_farm activities | Whether the household gets involved in off-farm income-generating activities | A dummy variable (1 = Yes and 0 = No) |
| Household Dietary Diversity | Different food groups are considered | Number |
| Livestock ownership (TLU) | Number of livestock a household owns | In Tropical Livestock Units using standard conversion factors |
| Access to credit | Whether the household has got access to credit or not | A dummy variable (1 = Yes and 0 = No) |

cut-off or threshold made it difficult to employ regression discontinuity analysis effectively. As our dataset was cross-sectional, the classic difference-in-difference model, which typically relies on pre- and post-treatment observations, was not applicable. Given these constraints, treatment effects models emerged as the most appropriate approach to estimate and understand the causal relationships in our study, allowing us to control for covariates and account for potential biases while assessing the impact of the treatment or intervention, i.e, productive safety net program.

## 2.2 Variable description

The variables used in this study include demographic, socio-economic and institutional factors. Table 1 describes these variables and how they are measured.

## 2.3 Characteristics of respondents

Table 2 provides descriptive statistics for several variables related to households. These variables include household size, age of the household head, gender of the household head,

**Table 2. Descriptive statistics.**

| Variables | mean | Std. dev | Min | Max |
|---|---|---|---|---|
| Household size | 5.89 | 1.96 | 2 | 10 |
| Age of the household head | 44.41 | 13.61 | 5 | 90 |
| Gender of the household head | 0.71 | 0.45 | 0 | 1 |
| Education level of the household head | 3.91 | 4.45 | 0 | 18 |
| Engagement in off_farm activities | 0.04 | 0.20 | 0 | 1 |
| Engagement in non_farm activities | 0.20 | 0.40 | 0 | 1 |
| Household Dietary Diversity | 3.51 | 1.85 | 0 | 8 |
| Livestock ownership (TLU) | 2.99 | 2.16 | 0 | 11.84 |
| Access to credit | 0.32 | 0.47 | 0 | 1 |

Source: Own analysis from the survey data

education level of the household head, engagement in off-farm activities, engagement in non-farm activities, household dietary diversity, livestock ownership (measured in TLU or Tropical Livestock Units), and access to credit. For household size, the average household in this dataset consists of approximately six people (with a standard deviation of 1.96.), with most households having between four and eight members. The minimum household size recorded is two, while the maximum is ten. The result also shows that the average household head is in their mid-forties (with a standard deviation of 13.61), but there is a significant amount of variation in age across the dataset. The minimum age recorded is 14, while the maximum is 90.

The mean is of the gender variable is 0.71, suggesting that the majority of household heads in this dataset are female. The education level of the household head is measured on a scale of 0–18, with higher scores indicating greater levels of education. The mean education level is 3.91, with a standard deviation of 4.45. This suggests that, on average, household heads in this dataset have completed some primary education, but there is a significant amount of variation in education levels.

Table 2 also includes information on whether households engage in off-farm and non-farm activities, with means of 0.04 and 0.20, respectively. Household dietary diversity is another variable listed, with a mean of 3.51 and a standard deviation of 1.85. The livestock ownership variable is measured in TLU (tropical livestock units) and has a mean of 2.99 and a standard deviation of 2.16. Finally, the table includes information on access to credit, with a mean of 0.32, indicating that about a third of households in the sample have access to credit.

## 3. Ethical consideration

The data collection methods and research tools employed in this study underwent a comprehensive review and received approval from Addis Ababa University as a crucial component of meeting the criteria for a PhD program. Subsequently, a letter granting ethical clearance has been issued and submitted to the journal for consideration.

Participants specifically the household heads involved in the study, were given the opportunity to provide their written consent, and they were also given the right to decline participation if they wished to do so. To ensure the privacy and confidentiality of the participants, all collected data was coded and did not include any identifying information. The researcher specifically did not collect any personal or identifying details.

The study adopted various measures to safeguard the well-being of participants. Specifically, participants received comprehensive briefings on the study's purpose and objectives, along with information about potential risks or adverse effects associated with their participation. The study explicitly ensured participants the right to withdraw at any point, with written consent seamlessly integrated into the questionnaire. This assurance instilled confidence in participants, providing them with the empowerment to withdraw from the study whenever they deemed it necessary.

Furthermore, the study took precautions to ensure the representativeness of the sample. A random selection process was employed to ensure that all potential participants had an equal chance of being included in the study, thereby promoting fair and unbiased representation.

Overall, the research was conducted with utmost care and attention to ensure the well-being and autonomy of the participants. Strict ethical guidelines were followed to avoid any harm, deception, or coercion. Participants were assured that the research findings would not be used to cause harm to them or others, but rather to gain insights into the effectiveness of the Productive Safety Net Program in Ethiopia.

**Table 3. Results of the independent samples test.**

| Variables | Membership in PSNP | Mean | Std. dev | Difference | t | p |
|---|---|---|---|---|---|---|
| Assets | Non_members | 0.297 | 0.130 | 0.068 | 4.5316 | 0.0000 |
| | Members | 0.228 | 0.132 | | | |
| Resilience | Non_members | 0.516 | 0.157 | 0.090 | 4.5501 | 0.0000 |
| | Members | 0.426 | 0.183 | | | |
| Log of income | Non_members | 5.757 | 0.067 | 0.304 | 3.0479 | 0.0025 |
| | Members | 5.453 | 0.074 | | | |

Source: Own analysis from the survey data

## 4. Results and discussion

### 4.1 Results of inferential statistics

Table 3 displays the results of an independent samples t-test comparing the means of members and non-members of the Productive Safety Net Program (PSNP) for three variables: Assets, Resilience, and Log of Income. For each variable, the table presents the mean and standard deviation for both members and non-members of the PSNP, as well as the difference between the means, the t-value, and the p-value. Overall, the results suggest that members of the PSNP have lower levels of assets and resilience, and significantly lower levels of income compared to non-members.

The results indicate that for all three variables, there is a significant difference between the means of members and non-members of the PSNP. For Assets, the mean for non-members is 0.297, with a standard deviation of 0.130, while the mean for members is 0.228, with a standard deviation of 0.132. The difference between the means is 0.068. The t-value is 4.5316, and the p-value is 0.0000, indicating that the difference is statistically significant at 1% level.

Similarly, for Resilience, the mean for non-members is 0.516, with a standard deviation of 0.157, while the mean for members is 0.426, with a standard deviation of 0.183. The difference between the means is 0.090. The t-value is 4.5501, and the p-value is 0.0000. For Income, the mean values are presented in logarithmic form. Accordingly, the mean for non-members is 5.757, with a standard deviation of 0.067, while the mean for members is 5.453, with a standard deviation of 0.074. The difference between the means is 0.304 or 30.4%. The t-value is 3.0479, with the p-value of 0.0025 showing significance at 1% level.

### 4.2 Results of the treatment effects models

The analysis of the Productive Safety Net Program (PSNP) in Ethiopia, utilizing various algorithms such as regression adjustment, inverse probability weighting, and propensity score matching, indicates no statistically significant impact on access to basic services (Table 4). Although all three algorithms show negative coefficients, suggesting a potential decrease in

**Table 4. Impact of PSNP on livelihood outcomes.**

| Algorithm | Access to Basic services | Income | Food Expenditure | Asset Ownership | Adaptive Capacity | Dietary Diversity |
|---|---|---|---|---|---|---|
| Regression Adjustment | -0.0005 (0.0107) | 0.3929 (0.5969) | -0.2964*** (0.1016) | -0.0626*** (0.0156) | -0.0065 (0.0225) | 0.0224 (0.2233) |
| Inverse Probability Weighting | -0.0008 (0.0108) | 0.4011 (0.5987) | -0.2976*** (0.1014) | -0.0624*** (0.0157) | -0.0071 (0.0226) | 0.0234 (0.2240) |
| Propensity score Matching | -0.0066 (0.0118) | 0.2540 (0.7022) | -0.3183*** (0.1164) | -0.0609*** (0.0175) | 0.0006 (0.0247) | -0.1165 (0.2440) |

Source: Own analysis from the survey data

access to basic services, none of these coefficients demonstrate statistical significance at conventional levels (p > 0.05). These findings align with similar studies, like [20, 21], emphasizing the PSNP's had no significant impact on food security or health outcomes and access to education or health services in Ethiopia, respectively. However, it is crucial to note that the absence of a significant impact on access to basic services does not necessarily imply the program's ineffectiveness. The PSNP might still have other vital impacts, such as poverty reduction or agricultural productivity enhancement, despite these findings.

Regarding its impact on income, the PSNP shows no significant effect across the applied algorithms, consistent with prior research indicating inconsistent or null effects on income in Ethiopia and other global contexts [22–24]. Despite observed improvements in food security and poverty reduction, studies highlight limited impacts on income or consumption among program participants. For example, a study by [23] found that the government's Productive Safety Net Program in Ethiopia had a positive impact on food security and asset accumulation, but no significant impact on income. Another study by [22] found that PSNP participants had lower levels of income and assets, but higher levels of food security and social capital compared to non-participants. A more recent study by [24] also found that the program had no significant impact on income or consumption, but did improve food security and reduce the risk of being in poverty. Similarly, a study by [25] on the same program found that it reduced poverty and increased food security, but had no significant impact on household income.

These inconsistent findings are not unique to Ethiopia. In sub-Saharan Africa, a study by [26] on social safety nets found that while cash transfer programs had positive effects on consumption and food security, they did not have a significant impact on income. Similarly, a study by [27] on the social cash transfer program in Zambia found that while the program improved the welfare of its beneficiaries, it had no significant impact on household income. A study by [1] on the effects of social protection programs in Africa also found that while these programs had positive impacts on health and education outcomes, they did not significantly increase income.

Globally, a study by [1] on social protection programs found that while cash transfers had positive effects on poverty reduction and human development outcomes, the evidence on their impact on income was inconclusive. Similarly, a study by [27] on the impact of social protection programs on income in low- and middle-income countries found that while these programs had positive impacts on poverty reduction and well-being, they did not necessarily increase income. A study by [28] on the impact of cash transfer programs in Asia and the Pacific found that while these programs reduced poverty and improved food security, they did not significantly increase income.

Examining food expenditure, all three models in Table 4 present negative and statistically significant coefficients (p < 0.05), suggesting that PSNP participation is linked to decreased food expenditure. These findings are consistent with previous studies that have also found a negative or null impact of social protection programs on food expenditure in Ethiopia, Africa, and globally. For example, a study by [25] found that the PSNP had no significant impact on food expenditure in Ethiopia, but did improve food security and reduce the risk of being in poverty. Another study by [27] found that the PSNP had no significant impact on food security or health outcomes in Ethiopia.

In sub-Saharan Africa, a study by [29] on social protection programs found that while cash transfer programs had positive effects on food consumption, they did not have a significant impact on food expenditure. Similarly, a study by [30] on social protection programs globally found that while cash transfers had positive effects on poverty reduction and human development outcomes, the evidence on their impact on food expenditure was inconclusive. The negative impact of the PSNP on food expenditure may reflect the challenges of designing effective social protection programs that address the multidimensional nature of poverty, particularly

in contexts of high food insecurity and limited access to basic services. It is possible that the PSNP is not providing sufficient support to its beneficiaries to maintain or increase their food expenditure, or that the program's implementation and targeting mechanisms are not reaching those who need it the most.

As depicted in Table 4, the negative and significant coefficients obtained from all the three treatment effect models suggest that the Productive Safety Net Program has a detrimental impact on asset ownership among member households. This means that the program leads to a reduction in assets owned by households that participate in the program. These results are consistent with findings from other studies on safety net programs in Ethiopia and Africa. For instance, a study by [31] found that Ethiopia's Productive Safety Net Program had a negative impact on asset accumulation, particularly for households that had been participating in the program for several years. Another study by [22] found that the program had a negative impact on asset accumulation and household welfare in Ethiopia's Tigray region. Another study by [22] found that PSNP participation was associated with lower levels of household assets compared to non-participants. These findings are also consistent with similar studies conducted globally. A review by [28] found that safety net programs in Africa, Asia, and Latin America often had negative effects on asset ownership and other dimensions of household welfare. There could be several reasons why safety net programs have a negative impact on asset ownership. For example, programs that provide cash transfers or food aid may discourage households from investing in income-generating activities or other long-term assets. Additionally, households that receive transfers may become dependent on the program, leading to a decline in their willingness to invest in productive assets.

In terms of adaptive capacity, the analysis in Table 4 indicates no significant effect of the PSNP across households. These findings are consistent with previous studies on the PSNP's impact on adaptive capacity. For example, a study by [32] found that the PSNP in Ethiopia had a limited impact on enhancing adaptive capacity, particularly in terms of diversifying livelihoods and increasing access to education and health services. Another study by [33] found that the PSNP had little impact on the adaptive capacity of rural households in Ethiopia, particularly in terms of reducing their vulnerability to shocks and stresses.

Similarly, a study by [34] found that PSNP participation was associated with lower levels of resilience in Ethiopia. These results are also consistent with similar studies on the impact of safety net programs on adaptive capacity in other regions of Africa and globally. For instance, a study by [35] found that a cash transfer program in Tanzania had limited impact on the adaptive capacity of households, particularly in terms of improving their access to education and health services. Another study by [28] similarly found that the Productive Inclusion Program (PIP) in Honduras had little impact on improving the adaptive capacity of poor households, particularly in terms of diversifying their income sources and reducing their exposure to shocks. This finding is also consistent with other studies that have found limited impact of safety net programs on adaptive capacity. For instance, a study by [36] found that social protection programs in Ethiopia and other African countries have had limited impact on the resilience of vulnerable households to climate change.

Regarding household dietary diversity, the study reveals no significant effect of the PSNP on dietary diversity in the study area. This finding is consistent with other studies on the PSNP's impact on household dietary diversity in Ethiopia. For example, a study by [12] found that the PSNP had no significant effect on the dietary diversity of rural households in northern Ethiopia. Similarly, a study by [37] found that the PSNP did not significantly improve the dietary diversity of rural households in southern Ethiopia. These results suggest that the PSNP may not be effectively addressing the underlying causes of food insecurity and malnutrition in the study area, such as poverty and limited access to diverse food sources.

Similar studies on the impact of safety net programs on dietary diversity in other regions of Africa and globally have also found mixed results. For example, a study by [13] found that cash transfers in Kenya led to significant improvements in dietary diversity among recipient households. However, a study by [21] found that a cash transfer program in Zambia had no significant impact on dietary diversity. These differences may be due to variations in the design and implementation of the programs, as well as differences in the socio-economic and cultural contexts in which they are implemented.

In a nutshell, while the PSNP demonstrates impacts in certain areas like food security and poverty reduction, the findings underscore challenges in achieving significant positive impacts across various dimensions of livelihoods. They emphasize the necessity for refined program designs and implementations to effectively address the multifaceted nature of poverty among vulnerable populations in Ethiopia.

## 5. Conclusions and policy implications

### 5.1 Conclusions

The article discusses the results of a study on the impact of the Productive Safety Net Program (PSNP) in Ethiopia on access to basic services, income, food expenditure, dietary diversity and adaptive capacity. The study used three different algorithms to analyze the data: regression adjustment, inverse probability weighting, and propensity score matching.

The results indicate that there is no statistically significant impact of the PSNP on access to basic services based on all three algorithms. The results also indicate that there is no significant impact of the PSNP on the income, dietary diversity and adaptive capacity of its members based on all three algorithms.

The results further indicate that there is a statistically significant negative impact of the PSNP on food expenditure and asset ownership based on all three algorithms. This suggests that the PSNP is associated with a decrease in food expenditure and assets suggesting that intermittent cash hand-outs are not adequate to cover expenses for food leading to asset run-downs, which is against the very objective PSNPs are designed. It is important to note that the null or mixed findings in this study may reflect the challenges of designing effective social protection programs that address the complex and multidimensional nature of poverty. The PSNP aims to provide both immediate relief and long-term investments in human capital and productive assets, which may require a longer timeframe and more nuanced measures of impact to assess fully. Additionally, the findings may be context-specific and may not generalize to other settings or programs. Further research is needed to identify effective approaches to social protection that can improve the income and well-being of vulnerable populations.

### 5.2 Policy implications

Based on the findings, we suggest that policy makers consider the following interventions:

1. Given the lack of significant impact on access to basic services and income, it may be necessary to reassess the design and implementation of the productive safety net program in Ethiopia to ensure that it effectively addresses the complex and multidimensional nature of poverty.

2. Consider the context-specific factors that may affect the impact of the program on the income and well-being of beneficiaries, including the socio-economic and political conditions of the beneficiaries and the implementation of the program.

3. To address the negative impact of the productive safety net program on food expenditure, policy-makers should consider complementary programs that focus on improving food

security, such as agricultural extension services, seed and fertilizer subsidies, and other nutrition interventions.

4. It may also be necessary to identify effective approaches to social protection that can improve the income and well-being of vulnerable populations, both in Ethiopia and globally, through further research and evaluation of existing programs.

5. Future studies may use more nuanced measures of impact and a longer timeframe to fully assess the effectiveness of the productive safety net program in Ethiopia.

## Supporting information

**S1 File. Questionnaire with written consent.**
(DOC)

**S2 File. Ethical approval from university department.**
(JPG)

## Author Contributions

**Writing – original draft:** Mesfin Getaneh Woldemichael, Bamlaku Alamirew Alemu.

**Writing – review & editing:** Mesfin Getaneh Woldemichael, Bamlaku Alamirew Alemu.

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
