## [Decision Letter · Decision Letter 0]

9 Jan 2024

PONE-D-23-32019Does Productive Safety Net Program Enhance Livelihoods? Insights from Vulnerable Households in Wolaita Zone, EthiopiaPLOS ONE

Dear Dr. Mesfin,

Thank you for submitting your manuscript to PLOS ONE. After careful consideration, we feel that it has merit but does not fully meet PLOS ONE’s publication criteria as it currently stands. Therefore, we invite you to submit a revised version of the manuscript that addresses the points raised during the review process.

**Comments from the reviewers are included at the end of this letter, and attention is to be paid to item 5 of the comments that talks about presentation of the findings and the refinement of the literature review section. **============================== Please submit your revised manuscript by Feb 23 2024 11:59PM. If you will need more time than this to complete your revisions, please reply to this message or contact the journal office at plosone@plos.org. Please include the following items when submitting your revised manuscript:A rebuttal letter that responds to each point raised by the academic editor and reviewer(s). You should upload this letter as a separate file labeled 'Response to Reviewers'.A marked-up copy of your manuscript that highlights changes made to the original version. You should upload this as a separate file labeled 'Revised Manuscript with Track Changes'.An unmarked version of your revised paper without tracked changes. You should upload this as a separate file labeled 'Manuscript'.If applicable, we recommend that you deposit your laboratory protocols in protocols.io to enhance the reproducibility of your results. Protocols.io assigns your protocol its own identifier (DOI) so that it can be cited independently in the future. For instructions see: https://journals.plos.org/plosone/s/submission-guidelines#loc-laboratory-protocols. Additionally, PLOS ONE offers an option for publishing peer-reviewed Lab Protocol articles, which describe protocols hosted on protocols.io. Read more information on sharing protocols at https://plos.org/protocols?utm_medium=editorial-email&utm_source=authorletters&utm_campaign=protocols.

We look forward to receiving your revised manuscript.

Kind regards,

Nkosiyazi Dube, Ph.D

Academic Editor

PLOS ONE

Journal Requirements:

Reviewers' comments:

Reviewer's Responses to Questions

**Comments to the Author**

1. Is the manuscript technically sound, and do the data support the conclusions?

Reviewer #1: Yes

2. Has the statistical analysis been performed appropriately and rigorously? 

Reviewer #1: I Don't Know

3. Have the authors made all data underlying the findings in their manuscript fully available?

Reviewer #1: No

4. Is the manuscript presented in an intelligible fashion and written in standard English?

Reviewer #1: Yes

5. Review Comments to the Author

Reviewer #1: This is generally a well written study. More work on representation of results could be beneficial to ensure readability. Similarly to ensure readability, literature review included in the background could be revised with more attention being towards the flow of this section.

I have included more extensive comments in the attached reviewed manuscript.

6. PLOS authors have the option to publish the peer review history of their article (what does this mean?). If published, this will include your full peer review and any attached files.

Reviewer #1: **Yes: **Malesedi P. Guambe

---

## [Author Response · Author response to Decision Letter 0]

10 Jan 2024

No Section Comments by Reviewers Responses

1 Abstract Highlight the study design and the software utilized : Done and included.

2 Abstract: key words Include variables of the study and remove Impact: Done and included 

3 Background Focus on each variable to each paragraph: Done and rearranged 

4 Data and methods Address same comment from abstract: Done and addressed

5 Data and methods Leave sentence which is unnecessary and lack specific: Done and removed

6 Data and methods Mention study area by its name: Changed and addressed

7 Data and methods Study variable could be presented as definition and question used to describe the variables , categories and codes, frequencies and percentages.: Done and included

8 Data and methods Provide the name of the formula in equation : Done and inserted

9 Data and methods Add as Appendix: addressed

10 Ethical consideration State whether you received ethics approval or waiver approval letter: Addressed, submitted and already got clearance form the journal

11 Ethical consideration Omission of repeated statement : Addressed

12 Ethical consideration State the measures to assure wellbeing of participants : Addressed and included 

13 Results and discussion Include the table 1 on methodology rather in results: Addressed and moved

14 Results and discussion Put the table 2 before the analysis.: Addressed and moved 

15 Results and discussion Create on table for all variables in one and include the analysis below. 

 :Addressed in one table and analyses followed

---

## [Editor Report · Decision Letter 1]

12 Jan 2024

Does Productive Safety Net Program Enhance Livelihoods? Insights from Vulnerable Households in Wolaita Zone, Ethiopia

PONE-D-23-32019R1

Dear Dr. MESFIN _ WOLDEMCHAEL,

We’re pleased to inform you that your manuscript has been judged scientifically suitable for publication and will be formally accepted for publication once it meets all outstanding technical requirements.

Kind regards,

Nkosiyazi Dube, Ph.D

Academic Editor

PLOS ONE

---

## [Editor Report · Acceptance letter]

15 Feb 2024

PONE-D-23-32019R1 

PLOS ONE

Dear Dr. WOLDEMICHAEL, 

I'm pleased to inform you that your manuscript has been deemed suitable for publication in PLOS ONE. Congratulations! Your manuscript is now being handed over to our production team.

Kind regards, 

on behalf of

Dr. Nkosiyazi Dube 

Academic Editor

PLOS ONE